# The *Chlamydia trachomatis* Inc Tri1 interacts with TRAF7 to displace native TRAF7 interacting partners

Clara M. Herrera,[1] Eleanor McMahon,[1] Danielle L. Swaney,[2] Jessica Sherry,[1] Khavong Pha,[1] Kathleen Adams-Boone,[1] Jeffrey R. Johnson,[2] Nevan J. Krogan,[2] Meredith Stevers,[3] David Solomon,[3] Cherilyn Elwell,[1] Joanne Engel[1,4]

**ABSTRACT** *Chlamydia trachomatis* is the leading cause of bacterial sexually transmitted infections in the USA and of preventable blindness worldwide. This obligate intracellular pathogen replicates within a membrane-bound inclusion, but how it acquires nutrients from the host while avoiding detection by the innate immune system is incompletely understood. *C. trachomatis* accomplishes this in part through the translocation of a unique set of effectors into the inclusion membrane, the inclusion membrane proteins (Incs). Incs are ideally positioned at the host-pathogen interface to reprogram host signaling by redirecting proteins or organelles to the inclusion. Using a combination of co-affinity purification, immunofluorescence confocal imaging, and proteomics, we characterize the interaction between an early-expressed Inc of unknown function, Tri1, and tumor necrosis factor receptor-associated factor 7 (TRAF7). TRAF7 is a multi-domain protein with a RING finger ubiquitin ligase domain and a C-terminal WD40 domain. TRAF7 regulates several innate immune signaling pathways associated with *C. trachomatis* infection and is mutated in a subset of tumors. We demonstrate that Tri1 and TRAF7 specifically interact during infection and that TRAF7 is recruited to the inclusion. We further show that the predicted coiled-coil domain of Tri1 is necessary to interact with the TRAF7 WD40 domain. Finally, we demonstrate that Tri1 displaces the native TRAF7 binding partners, mitogen-activated protein kinase kinase kinase 2 (MEKK2), and MEKK3. Together, our results suggest that by displacing TRAF7 native binding partners, Tri1 has the capacity to alter TRAF7 signaling during *C. trachomatis* infection.

**IMPORTANCE** *Chlamydia trachomatis* is the leading cause of bacterial sexually transmitted infections in the USA and preventable blindness worldwide. Although easily treated with antibiotics, the vast majority of infections are asymptomatic and therefore go untreated, leading to infertility and blindness. This obligate intracellular pathogen evades the immune response, which contributes to these outcomes. Here, we characterize the interaction between a *C. trachomatis*-secreted effector, Tri1, and a host protein involved in innate immune signaling, TRAF7. We identified host proteins that bind to TRAF7 and demonstrated that Tri1 can displace these proteins upon binding to TRAF7. Remarkably, the region of TRAF7 to which these host proteins bind is often mutated in a subset of human tumors. Our work suggests a mechanism by which Tri1 may alter TRAF7 signaling and has implications not only in the pathogenesis of *C. trachomatis* infections but also in understanding the role of TRAF7 in cancer.

**KEYWORDS** *Chlamydia trachomatis*, inclusion membrane protein, TRAF7, host-pathogen interaction, MEKK2, MEKK3, WD40, mass spectrometry

C*hlamydia trachomatis* is the most common cause of sexually transmitted bacterial infections and the leading cause of non-congenital blindness in developing nations (1). Complications associated with sexually transmitted serovars include ectopic

Address correspondence to Joanne Engel, joanne.engel@ucsf.edu, or Cherilyn Elwell, cherilyn.elwell@ucsf.edu.

The authors declare no conflict of interest.

See the funding table on p. 14.

10.1128/spectrum.00453-24   **1**

pregnancy, infertility, and pelvic inflammatory disease. Although effective antibiotic therapy exists, genital *C. trachomatis* infections are most commonly asymptomatic. Moreover, no therapy is cost-effective enough for use in developing nations, and an effective vaccine is still elusive (2). Elucidating the cellular and molecular mechanisms that contribute to *C. trachomatis* pathogenesis would allow the development of more targeted forms of treatment for the disease and could lead to the development of a vaccine.

*Chlamydia* species are obligate intracellular pathogens that share a common life cycle in which they alternate between the infectious extracellular elementary body (EB) and the intracellular replicative reticulate body (3). Upon binding to the host cell, EBs are internalized within a membrane-bound compartment through receptor-mediated endocytosis. *Chlamydia* avoids detection by the cytosolic innate immune response and escapes lysosomal degradation by modifying this compartment to create a unique intracellular replicative niche, the inclusion. *Chlamydia* species employ a specialized protein secretion apparatus, the type III secretion system, that translocates up to ~150 bacterial effectors into the host cytosol or into the inclusion membrane (4). This latter group, the inclusion membrane proteins (Incs), is unique to the order *Chlamydiales* (5). They are characterized by the presence of two or more closely linked predicted transmembrane domains with their N- and C-termini exposed to the host cell cytosol (6, 7). The roles of most *Chlamydia*-secreted effectors, including the Incs, remain incompletely described (8), as bioinformatics has been unrevealing, and conventional genetic manipulation of *Chlamydia* species has only recently been achieved (9). Based on their topology and functional studies, Incs may function as scaffolds to recruit and/or subvert host-cell proteins or organelles (3, 10–12).

To test this hypothesis and to identify host interacting partners that would give insights into Inc function, we previously performed a high throughput affinity purification-mass spectrometry (AP-MS) screen using epitope-tagged *C. trachomatis* serovar D Incs transfected into human cells (hereafter referred to as the "transfection interactome") (13). This screen identified potential binding partners for ~2/3 of the 58 predicted *C. trachomatis* serovar D Incs, including a predicted high confidence interaction between the ubiquitin ligase TRAF7 and CT224/CTL047 (hereafter renamed Tri1, for TRAF7 interactor). Here, we investigated the Tri1:TRAF7 interaction because Tri1 is not a well-characterized Inc, and because TRAF7 is involved in signaling pathways that are relevant for *Chlamydia* infection (14–19). Thus, decoding the Tri1:TRAF7 interaction may provide insights into *C. trachomatis* pathogenesis.

Tri1 is expressed early and throughout infection (20) and localizes to the inclusion membrane (21). TRAF7 is part of the tumor necrosis receptor-associated factor (TRAF) family of proteins, which use a modular structure to mediate the assembly of cytoplasmic signal transducers with regulator molecules downstream of receptor complexes (22). Each TRAF family member is thought to have distinct biological roles, even though TRAFs 1–6 regulate overlapping signaling pathways, including immune signaling (22, 23). Similarly to many of the other TRAFs, TRAF7 contains a coiled-coil domain important in multimerization (15, 24), a zinc (Zn) finger domain, and RING finger ubiquitin ligase domain (15, 16, 22, 25). However, unlike the other TRAFs, TRAF7 encodes a WD40 domain, a known protein scaffolding domain (26), in place of a canonical TRAF domain, at its C-terminus (22, 25).

In this work, we show that Tri1 and TRAF7 specifically interact during infection, and TRAF7 is recruited to the inclusion. We further show that the predicted coiled-coil domain of Tri1 is necessary and likely sufficient to interact with the TRAF7 WD40 domain. Finally, we demonstrate that Tri1 displaces native TRAF7-binding partners, mitogen-activated protein kinase kinase kinase 2 (MEKK2) and MEKK3. Together, our results suggest that Tri1 has the capacity to modulate TRAF7 protein-protein interactions (PPIs) and potentially alter TRAF7 signaling during infection.

## RESULTS

### TRAF7 co-affinity purifies with Tri1

Our previously published Inc-human transfection interactome predicted a high-confidence interaction between Tri1 (from *C. trachomatis* serovar D) transiently expressed in HEK293T cells and the human ubiquitin ligase TRAF7 (13). Given that *C. trachomatis* genetics is most well developed for serovar LGV434/L2 (9) and that Tri1 is highly conservedd between serovars D and L2 (98% identity, Fig. S1), we performed all subsequent studies with L2. We used two approaches to evaluate whether Tri1 interacts with TRAF7 in the context of L2 infection.

First, we performed affinity purifications (AP) from lysates of HeLa cells that were first transfected with mCherry (mCh)-TRAF7 for 24 h and then infected cells with L2 transformed with a plasmid expressing Tri1 ($Tri1_{FLAG}$) under the control of an anhydrous tetracycline (aTc)-inducible promoter ($L2+pTri1_{FLAG}$). As a control for Inc specificity, we infected cells in parallel with L2 transformed with a plasmid expressing $IncG_{FLAG}$ ($L2+pIncG_{FLAG}$), an unrelated Inc. Immunoblot analysis of eluates demonstrated that TRAF7 co-affinity purifies (co-APs) with $Tri1_{FLAG}$ but not with $IncG_{FLAG}$ (Fig. 1A).

Second, as the above experiment was performed with transfected TRAF7, we used AP-MS to determine whether endogenous TRAF7 co-APs with Tri1 during infection. Lysates from HeLa cells infected with $L2+pTri1_{FLAG}$ in the presence of inducer were affinity purified using FLAG beads, and the eluates were analyzed by liquid chromatography (LC)/MS-MS. As controls for specificity, we performed LC/MS-MS on eluates prepared in parallel from cells infected with L2+pVector or from cells infected with L2 strains expressing Incs for which host binding partners are established, IncE (10, 13, 27, 28) ($L2+pIncE_{FLAG}$) and Dre1 (13, 29) ($L2+pDre1_{FLAG}$). Protein-protein interactions were scored by SAINT (30), which uses a scale of 0–1. The SAINT score of 1 for the $Tri1_{FLAG}$:TRAF7 interaction with a Bayesian false discovery rate (BFDR) of 0 indicates a high confidence and specific interaction. It compares favorably with the SAINT score from previously validated interactions of $IncE_{FLAG}$ with SNX5 (10, 13, 27, 28) and of $Dre1_{FLAG}$ with DCTN4 (13, 29 ). No spectral counts were detected for TRAF7 in the eluates analyzed from $IncE_{FLAG}$ or $Dre1_{FLAG}$ (Fig. 1B), demonstrating the specificity of TRAF7 binding to Tri1. Likewise, no other TRAFs or any other WD40-domain-containing proteins were found to co-AP with Tri1. We thus conclude that in the context of transfection and infection, the interaction between Tri1 and TRAF7 is specific.

### TRAF7 is recruited to the inclusion

Having confirmed that Tri1 and TRAF7 interact during infection, we tested the prediction that Tri1 would recruit TRAF7 to the inclusion. We confirmed that $Tri1_{FLAG}$ localized to the inclusion membrane (Fig. 2A and B), consistent with the published results of others (21). We next examined the localization of mCh-TRAF7 and endogenous TRAF7 in cells infected with $L2+pTri1_{FLAG}$. We observed that mCh-TRAF7 was robustly recruited to the inclusion in the presence of inducer (Fig. 2A). We observed occasional recruitment of mCh-TRAF7 to the inclusions in infections performed in the absence of inducer (Fig. 2A), likely due to chromosomally expressed Tri1.

To examine the recruitment of endogenous TRAF7, we first validated a commercially available antibody to TRAF7 by immunofluorescence and immunoblot analysis (Fig. S2). Both endogenous TRAF7 and FLAG-TRAF7 could be detected by immunoblot (Fig. S2A). We observed that endogenous TRAF7 and mCh-TRAF7 localized to the nucleus, cytosol, and cytoplasmic puncta (Fig. S2B and C), consistent with previously published results (14, 15, 31). The cytoplasmic puncta were particularly prominent upon transfection, suggesting that TRAF7 puncta formation is enhanced when the protein is overexpressed, as has been reported for all of the TRAFs (32). In HeLa cells infected with $L2+pTri1_{FLAG}$ for 24 h, we observed recruitment of endogenous TRAF7 to the inclusion in the presence of inducer (Fig. 2B). Although recruitment of endogenous TRAF7 appears to be enhanced

**FIG 1** Tri1 and TRAF7 interact during infection. (A) HeLa cells were transiently transfected with mCh-TRAF7 for 24 h and then infected with L2+pIncG$_{FLAG}$ or L2+pTri1$_{FLAG}$ for 24 h in the presence of inducer (aTc). Shown are the lysates and FLAG-bead affinity purified eluates immunoblotted with the indicated antibodies. Glyceraldehyde phosphate dehydrogenase (GAPDH)) serves as a loading control, and *Chlamydia* major outer membrane protein (MOMP) serves as a control for the efficiency of infection. (B) FLAG-bead affinity purified eluates prepared from HeLa cells infected with L2 expressing the indicated Incs (pTri1$_{FLAG}$, pIncE$_{FLAG}$, or pDre1$_{FLAG}$) in the presence of inducer (aTc) were analyzed by liquid chromatography (LC)/MS-MS. Shown are selected average spectral counts from biological triplicates, SAINT scores, and Bayesian false discovery rate (BFDR). SAINT scores closer to 1 with a BFDR ≤ 0.05 suggest a high confidence interaction. N/D, not determined because no spectral counts were recorded. SNX5, sorting nexin 5. DCTN4, dynactin 4.

when Tri1 is overexpressed, it was not possible to quantify due to the low levels of detection of endogenous TRAF7 by the antibody.

## The predicted coiled-coil domain of Tri1 interacts with the WD40 domain of TRAF7

We next defined the regions of Tri1 and TRAF7 that are necessary and sufficient for their interaction by co-transfection followed by co-AP studies using deletion mutants (Fig. 3 and 4). Tri1 has a canonical Inc structure (21), with two closely linked transmembrane domains, a short N-terminal domain predicted to encode the type 3 secretion signal and a C-terminal domain (Fig. 3A).

Although alpha-fold analysis (33) predicts that the C-terminus would assume an alpha-helical structure, multiple coiled-coil prediction algorithms (34) suggest that the C-terminus encodes a coiled-coil domain. Based on these analyses, we assigned residues 118–138 for the coiled-coil domain of Tri1.

To determine if the coiled-coiled domain was necessary for Tri1 binding to TRAF7, we co-transfected HEK293T cells with full-length FLAG-TRAF7 and with Strep-tagged constructs of Tri1 that either completely lack the coiled-coil domain (Tri1$_{1-117}$, Fig. 3A) or that have a truncated coiled-coil domain (Tri1$_{1-128}$, Fig. 3A). To determine if the coiled-coil domain was sufficient for Tri1 binding to TRAF7, we fused Tri1$_{84-147}$ to superfolder GFP (sfGFP), in addition to Strep (Fig. 3A), to enhance its stability. Tri1-Strep or Tri1-sfGFP-Strep served as full-length controls. Lysates were affinity purified on Strep-Tactin beads. TRAF7 co-AP'd only with Tri1 variants containing the complete predicted coil-coil domain (Tri1 and Tri1$_{84-147}$; Fig. 3B), indicating that the coiled-coil domain is necessary and likely sufficient for the Tri1:TRAF7 interaction.

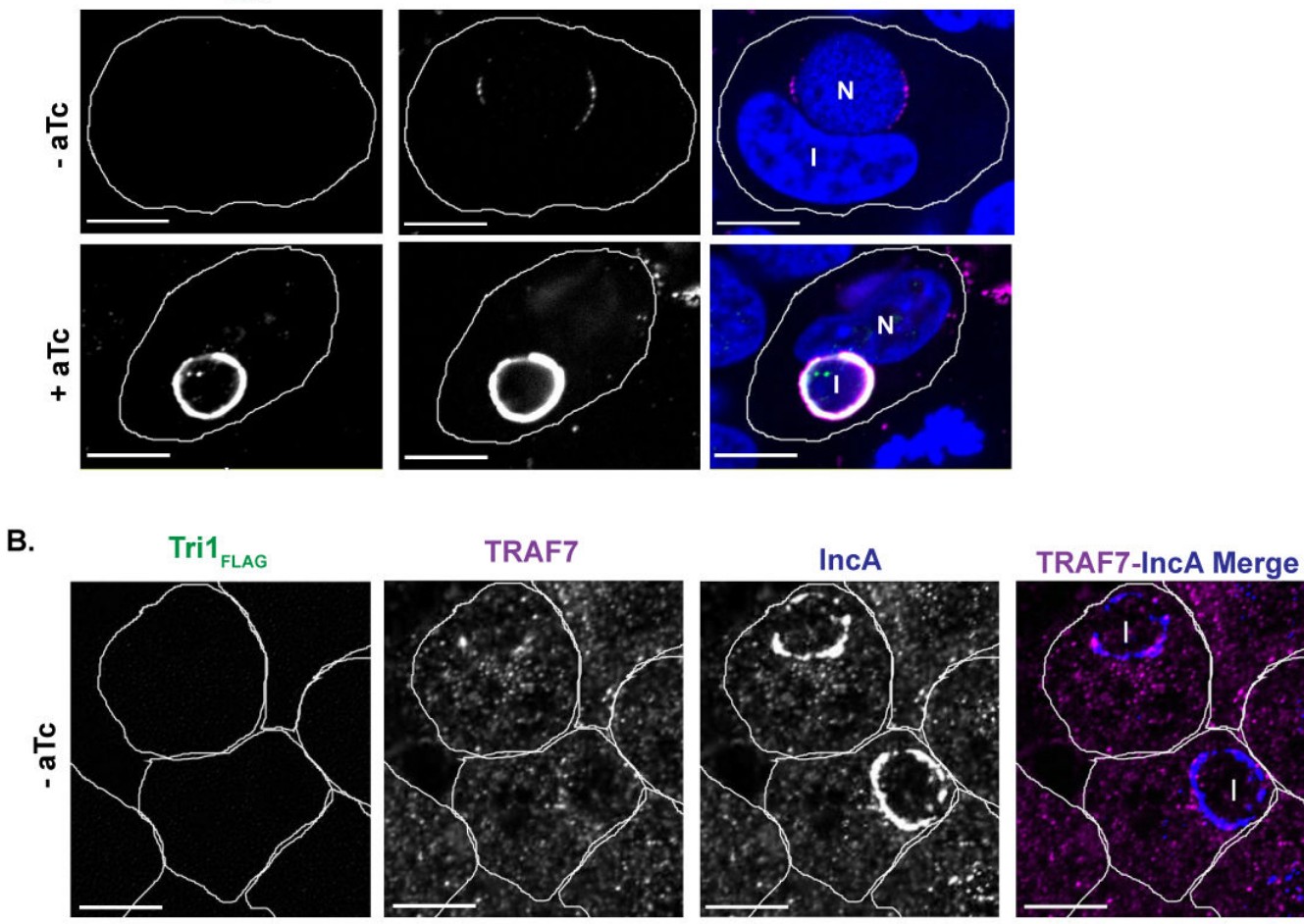

**FIG 2** TRAF7 is recruited to the inclusion. (A) Confocal immunofluorescence microscopy of HeLa cells transfected with mCh-TRAF7 for 24 h and then infected with L2+pTri1$_{FLAG}$ for 24 h with or without aTc induction. Cells were fixed and stained with α-FLAG to visualize Tri1$_{FLAG}$. Merged images show mCh-TRAF7 (pseudo-colored magenta), Tri1$_{FLAG}$ (pseudo-colored green), and DAPI (4′,6-diamidino-2-phenylindole) (blue) staining. (B) Confocal immunofluorescence microscopy of HeLa cells infected with L2+pTri1$_{FLAG}$ for 24 h with or without aTc induction. Cells were fixed and stained with antibodies to FLAG (to detect Tri1), TRAF7 (pseudo-colored magenta in merge), and IncA (blue in merge, to delineate the inclusion membrane). The merge panel only includes TRAF7 and IncA. The small amount of transfected mCh-TRAF7 present at the inclusion in the absence of inducer likely represents recruitment by chromosomally encoded Tri1. Shown are single z-slices. I, inclusion. N, nucleus. Scale bar = 10 µm. Cell outlines are included for clarity.

TRAF7 encodes several annotated domains (15, 16, 25), including an N-terminal RING finger ubiquitin ligase domain, a zinc (Zn) finger domain, a coiled-coil domain, and seven WD40 repeats that form the WD40 domain (Fig. 4A). We predicted that the interaction between TRAF7 and Tri1 is likely mediated by the WD40 domain, as only TRAF7, among the TRAF family members, encodes a WD40 domain (22), and no other proteins

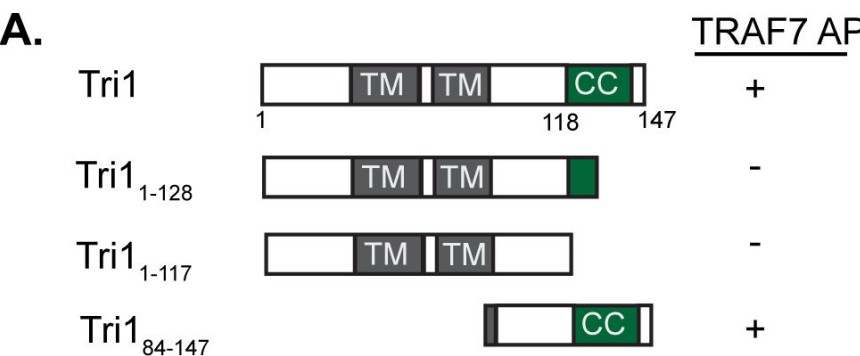

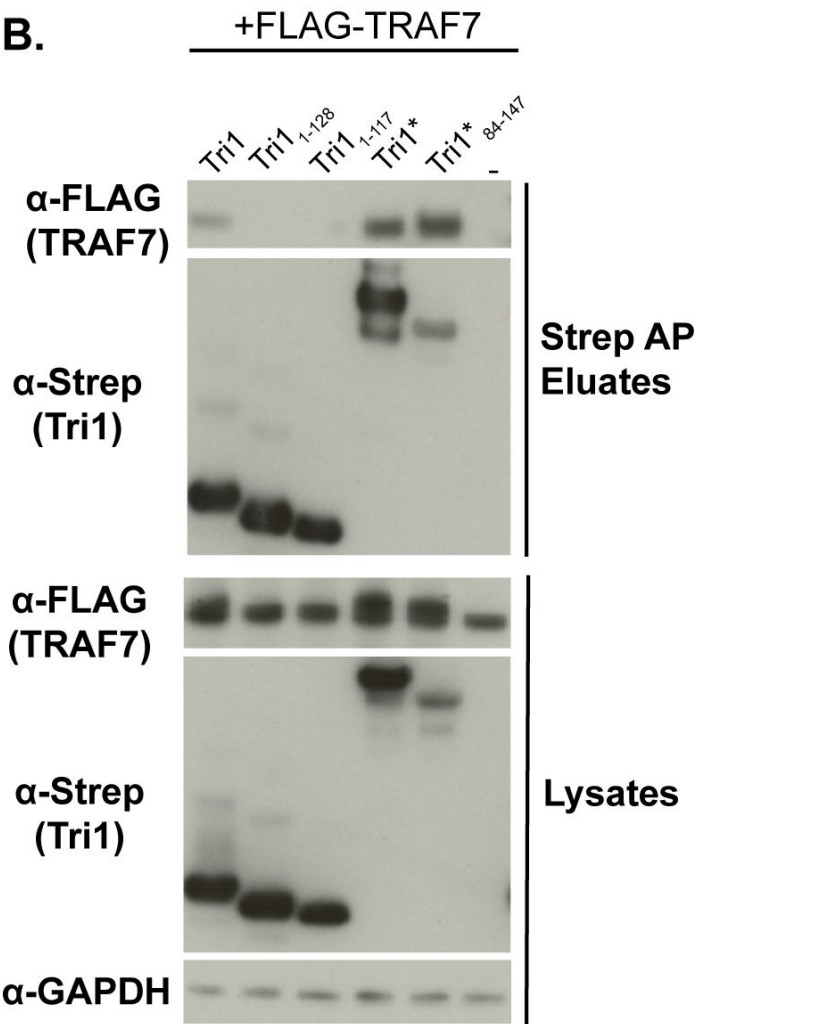

**FIG 3** The coiled-coil domain of Tri1 interacts with TRAF7. (A) Schematic of Strep-tagged Tri1 variants. Variants that interact with TRAF7 by co-AP are indicated with a "+" sign, and variants that do not interact are indicated with a "−" sign. (B) Lysates and eluates from affinity purifications of HEK293T cells co-transfected with the indicated Tri1-Strep or Strep-sfGFP-tagged variants (indicated with *) and with FLAG-TRAF7 were immunoblotted with the indicated antibodies. The control condition in which cells were transfected only with FLAG-TRAF7 is designated "−." GAPDH serves as a loading control for the lysates. Only Tri1 variants containing a complete coiled-coil domain (Tri1-Strep, Tri1- Strep-sfGFP, and Tri1$_{84-147}$- Strep-sfGFP) co-AP'd with TRAF7.

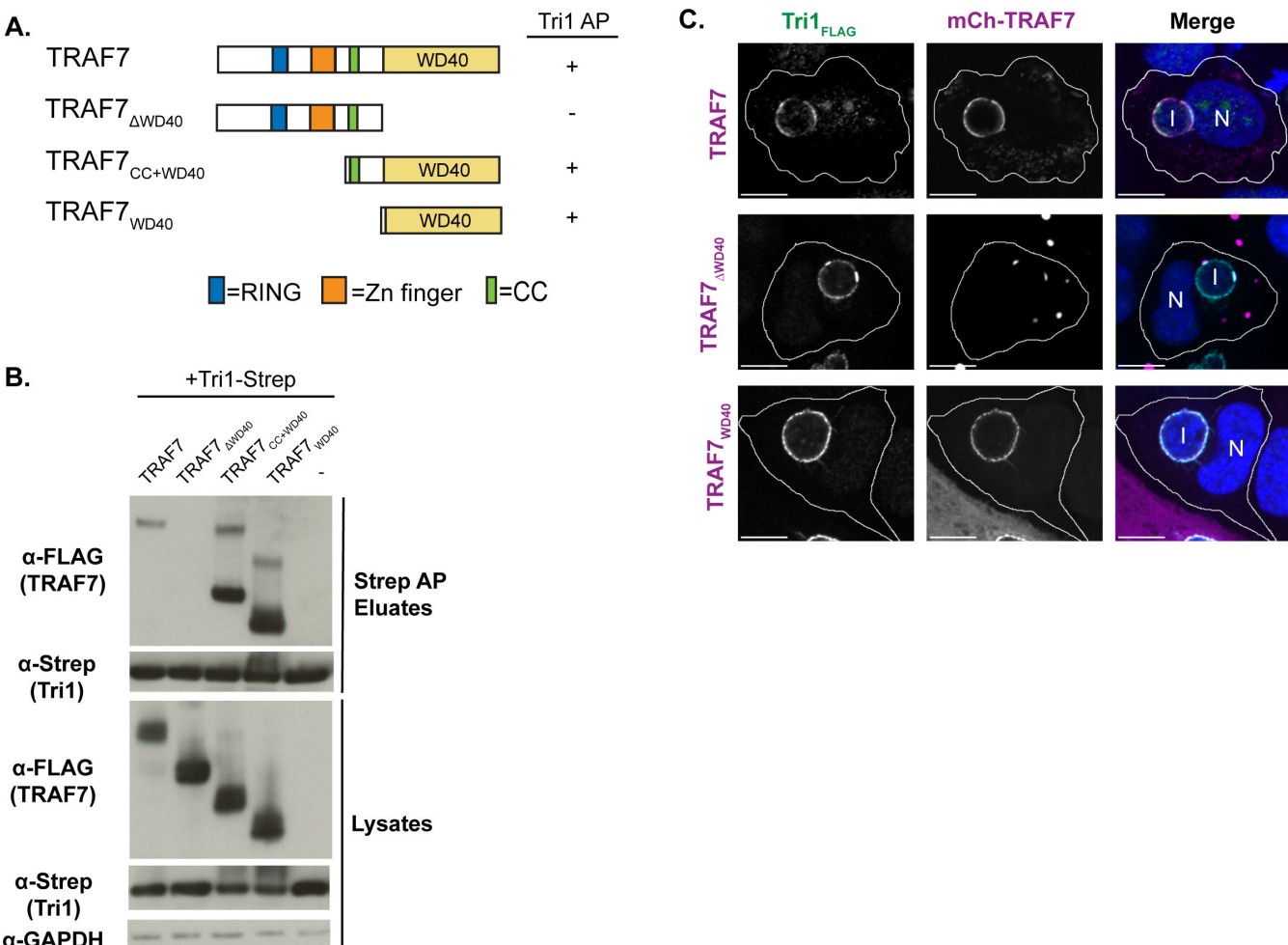

**FIG 4** The WD40 of TRAF7 is necessary and sufficient to interact with Tri1. (A) Schematic of TRAF7 constructs. Zn, zinc. CC, coiled-coil. RING, RING finger ubiquitin ligase domain. Variants that interact with TRAF7 by co-AP are indicated with a "+" sign, and variants that do not interact are indicated with a "−" sign. (B) Lysates and eluates of HEK293T cells co-transfected with Tri1-Strep and the indicated FLAG-TRAF7 variants ("−" indicates the control with no TRAF7 added) were affinity purified with Strep-Tactin beads and immunoblotted with the indicated antibodies. GAPDH serves as a loading control for lysates. Only variants containing the WD40 domain of TRAF7 co-affinity purified with Tri1. The slower migrating band present in some of the TRAF7 samples likely represents stable dimers. (C) Confocal immunofluorescence microscopy of HeLa cells transfected with the indicated mCh-TRAF7 variants for 24 h followed by infection L2+pTri1$_{FLAG}$ in the presence of aTc for 24 h. Cells were fixed and stained with α-FLAG and DAPI and imaged by confocal microscopy. Cell membranes are outlined. Tri1 is pseudocolored green and TRAF7 is pseudocolored magenta in the merged image. Shown are single z-slices. Scale bar = 10 µm. I, inclusion. N, nucleus.

containing a WD40 domain were found to co-AP with Tri1. To test this hypothesis, we performed co-APs on lysates prepared from HEK293T cells transfected with Tri1-Strep and with TRAF7 variants. Only TRAF7 variants containing the WD40 domain (TRAF7, TRAF7$_{CC+WD40}$, and TRAF7$_{WD40}$) co-AP'd with Tri1 (Fig. 4B). We thus conclude that the TRAF7 WD40 domain is necessary and sufficient for binding to Tri1.

We additionally tested whether the WD40 domain of TRAF7 was necessary or sufficient for its recruitment to the inclusion. We observed that mCh-TRAF7$_{WD40}$ and full-length mCh-TRAF7 were recruited to the inclusions of cells infected with L2+pTri1$_{FLAG}$ in the presence of inducer. In contrast, the TRAF7 variant that lacks the WD40 domain (mCh-TRAF7$_{\Delta WD40}$) exhibited minimal recruitment (Fig. 4C). We speculate that this residual recruitment to the inclusion is likely due to the ability of mCh-TRAF7$_{\Delta WD40}$ to multimerize with endogenous TRAF7 (15, 24).

## Tri1 displaces native TRAF7 interacting partners MEKK2 and MEKK3

We considered whether the binding of Tri1 to TRAF7$_{WD40}$ domain could disrupt native TRAF7 protein-protein interactions. For this purpose, we employed an unbiased quantitative AP-MS approach to identify native binding partners of the WD40 domain of TRAF7 whose interactions are disrupted in the presence of the coiled-coil domain of Tri1 but not by a Tri1 variant that lacks this domain (Fig. 5A). Cells co-transfected with empty FLAG-vector and Tri1-Strep served as negative controls. SAINT analysis identified MEKK2 as a significant interactor of TRAF7$_{WD40}$ in the presence of the truncated Tri1 (Tri1$_{1-128}$), with a SAINT score of 1 and a BFDR of 0 (Fig. 5B). However, in a parallel experiment in which full-length Tri1 was co-transfected with TRAF7$_{WD40}$, no spectral counts for MEKK2 were identified by AP-MS (Fig. 5B). This analysis suggests that MEKK2 is a native interactor of the WD40 domain of TRAF7. Furthermore, our results demonstrate that

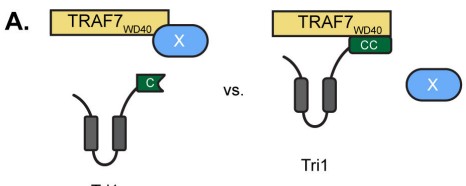

**B.**

| Bait | Condition | Prey | Avg. Spectral Counts | SAINT | BFDR |
|---|---|---|---|---|---|
| TRAF7$_{WD40}$ | Tri1$_{1-128}$ | MEKK2 | 7.7 | 1.0 | 0.00 |
| TRAF7$_{WD40}$ | Tri1 | MEKK2 | 0.0 | 0.0 | 0.58 |
| TRAF7$_{WD40}$ | Tri1$_{1-128}$ | TCPD | 18.0 | 1.0 | 0.00 |
| TRAF7$_{WD40}$ | Tri1 | TCPD | 19.7 | 1.0 | 0.00 |
| TRAF7$_{WD40}$ | Tri1$_{1-128}$ | TCPG | 30.3 | 1.0 | 0.00 |
| TRAF7$_{WD40}$ | Tri1 | TCPG | 30.7 | 1.0 | 0.00 |
| TRAF7$_{WD40}$ | Tri1$_{1-128}$ | DNJA2 | 7.0 | 1.0 | 0.00 |
| TRAF7$_{WD40}$ | Tri1 | DNJA2 | 8.3 | 1.0 | 0.00 |

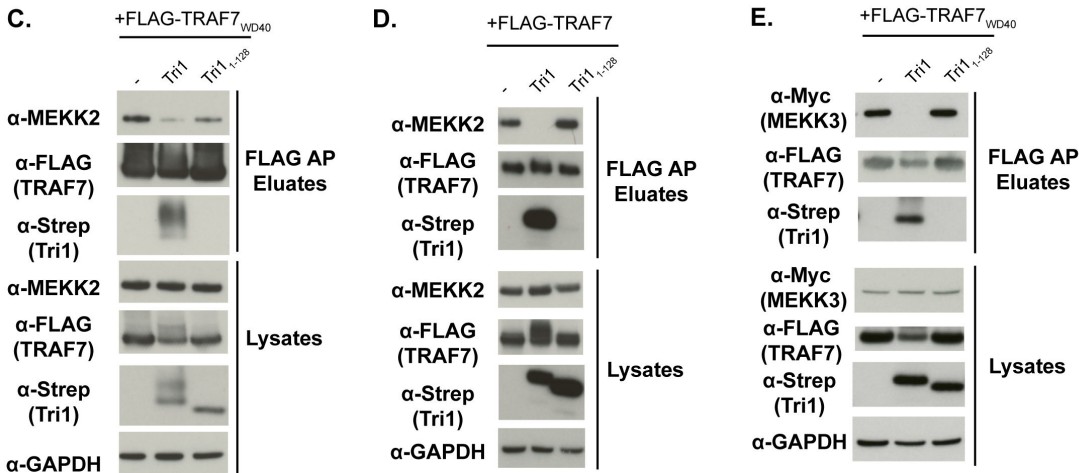

**FIG 5** Tri1 displaces MEKK2 and MEKK3 binding to TRAF7. (A) Schematic of displacement AP-MS analysis with a potential displaced TRAF7 native interactor represented by "X." (B) HEK293T cells co-transfected with FLAG-TRAF7$_{WD40}$ (bait) and either Tri1$_{1-128}$-Strep or Tri1-Strep. Lysates were affinity purified over FLAG beads and analyzed by LC/MS-MS. Shown are the average spectral counts from three biological replicates, SAINT scores, and BFDR of selected TRAF7$_{WD40}$ interacting partners in the presence of Tri1$_{1-128}$-Strep or Tri1-Strep. SAINT scores closer to 1 with a BFDR ≤ 0.05 suggest a high confidence interaction. Tri1 displaces MEKK2 binding to TRAF7$_{WD40}$ but not to TCPD, TCPG, or DNJA2. (C–E) Validation of Tri1-mediated displacement of MEKK2 and MEKK3 binding to the TRAF7 WD40 domain by co-transfection studies. HEK293T cells were co-transfected with either Tri1-Strep or Tri1$_{84-147}$- Strep (C–E), FLAG-TRAF7$_{WD40}$ (C and E), FLAG-TRAF7 (D), and Myc-MEKK3 (E). Cells only transfected with FLAG-TRAF7$_{WD40}$ (C and E) and FLAG-TRAF7 (D) were designated "−" and served as a control. Lysates were affinity purified using FLAG beads and analyzed by immunoblot with the indicated antibodies. GAPDH serves as a loading control for the lysates. Full-length Tri1, but not Tri1$_{1-128}$, disrupts TRAF7 binding to MEKK2 and to MEKK3.

Tri1 displaces MEKK2 from TRAF7 in a manner dependent upon the complete coil-coil domain of Tri1. We identified other interactors of the TRAF7 WD40 domain, including T-complex protein 1 subunit subunits delta and gamma (TCPD and TCPG) and DnaJ homolog subfamily A member 2 (DNJA2), which bound to TRAF7 in the presence of either Tri1 or Tri1$_{1-128}$ (Fig. 5B), indicating the Tri1-mediated displacement of MEKK2 is specific.

MEKK3, a closely related homolog of MEKK2, has also been reported to interact with the WD40 domain of TRAF7 (24, 26). However, we did not detect MEKK3 in our AP-MS data set for unclear reasons. We, therefore, tested whether Tri1 can disrupt MEKK2 or MEKK3 binding from the TRAF7 WD40 domain by co-transfection studies. HEK293T cells were co-transfected with Tri1-Strep or Tri1$_{1-128}$-Strep and either FLAG-TRAF7$_{WD40}$ (Fig. 5C and E) or FLAG-TRAF7 (Fig. 5D). For the MEKK3 displacement experiments, we additionally co-transfected cells with Myc-MEKK3 (Fig. 5E). Lysates were affinity purified over FLAG beads, and eluates were analyzed by immunoblot. Tri1, but not the non-binding variant (Tri1$_{1-128}$), decreased the binding of MEKK2 to both TRAF7$_{WD40}$ (Fig. 5C) and to TRAF7 (Fig. 5D). Similarly, Tri1, but not the non-binding variant (Tri1$_{1-128}$), decreased the binding of myc-MEKK3 to TRAF7$_{WD40}$ (Fig. 5E). Together, our studies demonstrate that Tri1, through its coiled-coil domain, can displace the TRAF7 native binding partners MEKK2 and MEKK3.

## DISCUSSION

As a successful intracellular pathogen, *Chlamydia* must replicate intracellularly while avoiding immune detection (3, 12, 35). Although *C. trachomatis* induces cell-autonomous signaling pathways, including the cGAS-STING, NF-kB, and JNK-AP-1 pathways, it must modulate their downstream effects to survive intracellularly (3, 12, 36). *C. trachomatis* is thought to regulate some of these pathways through the secretion of soluble effectors, including Incs (3, 11, 12). For example, upon transfection, the secreted effector Cdu1, a deubiquitinase and acetyltransferase (37), can inhibit NF-kB signaling (38). Additionally, the Inc CpoS can inhibit STING activation, possibly through its interaction with a subset of Rab GTPases (39, 40). However, no *C. trachomatis* Incs have as yet been linked to JNK-AP1 signaling, a pathway that is induced late during infection and is important for its intracellular replication (36). There is a precedent for inhibition of immune signaling in another human-adapted *Chlamydia* species, *C. pneumoniae*. Here, the Inc CP036 binds to and redirects the interleukin 17A receptor-binding protein Act1 to the inclusion, modulating Interleukin 17A signaling (41). Thus, further study into *C. trachomatis* effectors may elucidate how this intracellular bacterium can subvert major host signaling pathways.

Here, building on our previous work in which we defined the Inc-host interactome (13), we investigate the predicted interaction between Tri1 and TRAF7. We were particularly interested in this interaction because TRAF7 modulates key components of the innate immune response, including Type I interferon (IFN) (17, 18), NF-kB, and JNK-AP-1 signaling (14–16). We demonstrate that Tri1 interacts with TRAF7 during infection, and it can recruit endogenous or transfected TRAF7 to the inclusion. We localized the Tri1:TRAF7 binding interface to the coiled-coil domain of Tri1 and the WD40 domain of TRAF7. Using AP-MS, we identified MEKK2 as a native TRAF7 interactor. Finally, through co-transfection studies, we confirmed that Tri1 binding to the WD40 domain of TRAF7 can displace MEKK2 as well as a closely related previously identified TRAF7 interactor, MEKK3 (15, 16). While our experiments were in progress, others reported an interaction between TRAF7 and MEKK2, along with MEKK3 and MEK5 (42). Together, these experiments underscore that the TRAF7 WD40 domain can interact with a subset of constituents of the MAP kinase pathways.

By binding to host proteins, Incs have the potential to disrupt native host-protein interactions and/or to synergize with host-binding partners to create new protein-binding surfaces (3, 10–12). In our studies, we used an unbiased AP-MS-based strategy to define native binding partners of a host protein of interest that can be targeted by a

microbial effector. This approach allowed us to identify a new TRAF7 interactor (MEKK2) that could be displaced by an Inc. A major advantage of this approach is that it allows for the identification of host protein-binding partners that can be displaced by an effector without knowing the specific residues required for the effector:host protein interaction. Moreover, this approach is equally applicable if the binding of an effector to a host target creates a new binding interface that recruits additional host proteins. Finally, the approach is easily scalable and widely applicable to any microbial effector, as demonstrated by our initial AP-MS screen to define the Inc-host interactome (13).

A major conundrum of TRAF7 biology is its roles in a wide array of signaling pathways, sometimes with reportedly contradictory outcomes. For example, TRAF7 has been shown to both promote or inhibit inflammation (22) by either activating (15, 16) or inhibiting (14) the NF-kB signaling pathway. Additionally, TRAF7-dependent ubiquitination can inhibit type I IFN signaling (17, 18) and promote epithelial-mesenchymal transition (19), a reprogramming event that is induced by *C. trachomatis* infection (43). Thus, further study of the consequences of the Tri1:TRAF7 interaction may provide new insights into TRAF7 modulation of these important signaling pathways.

The WD40 domain of TRAF7 has recently emerged as a common mutational hot spot for a subset of unusual human tumors, including mesotheliomas (44, 45), adenomatoid tumors of the genital tract (46), fibromyxoid spindle-cell sarcomas of soft tissue (47), perineurioma nerve sheath tumors (48), and meningiomas (49–51). Some of the mutations lead to increased NF-kB signaling (46) and decreased JNK signaling(42). Particularly relevant to our studies of the Tri1:TRAF7 interaction are mutations within the TRAF7 WD40 domain associated with meningiomas. Transfection of these mutated proteins leads to decreased binding of MEKK3 to TRAF7 and decreased JNK signaling (42). These mutations predict potential residues that mediate the TRAF7:Tri1 interaction. Experiments to test this idea are underway.

Although data regarding the consequences of the TRAF7:MEKK2 interaction is limited, the role of TRAF7 in MEKK3 signaling is more well characterized (15, 16, 42). Therefore, we can speculate on how Tri1 displacement of MEKK3 from TRAF7 could modify host cell signaling. MEKK3 is a key signaling molecule in the TNF-induced NF-kB activation pathway (52). TRAF7 can stimulate MEKK3 kinase activity (16), and MEKK3 can promote TRAF7 ubiquitin ligase activity (15). Together, the two proteins act synergistically, and possibly in the same complex, to stimulate the NF-kB and JNK signaling pathways (15, 16 ). Additionally, TRAF7 is important for activating the MEKK3-MEK-ERK5 signaling pathway, which is involved in maintaining vascular cell integrity in response to fluid shear stress (42). This pathway has major consequences for cells, including inhibiting apoptosis, inflammation, and epithelial-to-mesenchymal transition (53), all of which could be relevant to *C. trachomatis* infection (43). We predict that Tri1 displacement of MEKK2/3 from TRAF7 during *C. trachomatis* infection would lead to a downregulation of TRAF7 ubiquitin ligase activity and thus alter TRAF7-dependent signaling, such as NF-kB, JNK, and MEKK3-MEK-ERK signaling (14–16).

Our identification of Tri1 as a TRAF7 interactor provides an excellent opportunity to tease apart the mechanistic functions of this important host protein in numerous signaling pathways related to developmental biology, immunology, and cancer biology. In this work, we focused on the TRAF7:MEKK2/3 complexes, but there are other TRAF7 protein-protein interactions that may be altered by Tri1 during infection. Thus, further investigation may identify additional native TRAF7 binding partners that are displaced by Tri1. Finally, it will be interesting to determine whether TRAF7 ubiquitin ligase activity and its related pathways are altered in a Tri1-dependent manner during infection. Examining these interactions will be key to fully understanding the role of Tri1 binding and recruitment of TRAF7 in *C. trachomatis* intracellular survival.

## MATERIALS AND METHODS

### Cell culture and bacterial propagation

HeLa 229 and Vero cells were obtained from the American Type Culture Collection (ATCC). HEK293T cells were a generous gift from Dr. Nevan Krogan (UCSF). TRAF7 isoform 1 CRISPR/Cas9 KO HeLa cells (Isoform 2 of TRAF7, which lacks residues 1–76, was not targeted by CRISPR/Cas9 and was thus still expressed in cells) were obtained from Abcam (ab264998, discontinued).

HeLa cells were cultured and maintained in Eagle's minimum essential medium (MEM; UCSF Cell Culture Facility) supplemented with 10% (vol/vol) fetal bovine serum (FBS) from Gemini at 37°C in 5% $CO_2$. HEK293T and Vero cells were cultured and maintained in Dulbecco's modified Eagle's medium (DMEM, UCSF Cell Culture Facility) supplemented with 10% (vol/vol) FBS at 37°C in 5% $CO_2$. Cells were routinely tested for mycoplasma (Molecular Probes, M-7006). *C. trachomatis* was routinely propagated in Vero cell monolayers as previously described (54). HeLa cells were used for all infection studies. HeLa and HEK293T cells were used for ectopic expression experiments. Stellar chemically competent *Escherichia coli* (Takara Bio) was used to prepare plasmids for ectopic expression in mammalian cells.

### Antibodies and reagents

Primary antibodies were obtained from the following sources: rabbit anti-RFP (Rockland, 600–401-379-RTU), mouse anti-FLAG (Millipore, F3165), rabbit anti-FLAG (Millipore, F7425), mouse anti-GAPDH (Millipore, MAB374), goat anti-MOMP L2 (Fitzgerald, 20C-CR2104GP), rabbit anti-TRAF7 (Novus Biologicals, NBP2-93316), rabbit anti-Strep TagII HRP (Novagen, 71591–3), rabbit anti-MEKK2 (Abcam, ab33918), and mouse anti-Myc (Thermo Fisher, R950-25). Mouse anti-IncA was kindly provided by Dr. Dan Rockey (Oregon State University). Secondary antibodies for immunofluorescence microscopy were derived from donkey and purchased from Life Technologies or Abcam: anti-rabbit Alexafluor 568, anti-rabbit Alexafluor 488, anti-mouse Alexafluor 405, and anti-goat Alexafluor 647.

A plasmid expressing $Tri1_{84-147}$-GFP (serovar D) was a generous gift from Dr. Raphael Valdivia (Duke University). Myc-MEKK3 (K391A and pcDNA3.1) was originally provided by Dr. Xin Lin (Addgene plasmid # 44157). *C. trachomatis* L2 (434/Bu) was a generous gift from Deborah Dean (UCSF). The *C. trachomatis* L2 strains overexpressing $Tri1_{FLAG}$ (originally "CT224-FLAG") and $Dre1_{FLAG}$ were generous gifts from Drs. Mary Weber (University of Iowa) and Ted Hackstadt (Rocky Mountain Laboratories). The *C. trachomatis* L2 strain overexpressing $IncE_{FLAG}$ and $IncG_{FLAG}$ were generous gifts from Dr. Isabelle Derré (University of Virginia).

Primers (Table S1) were commercially generated by Integrated DNA Technologies or by Elim Biopharm.

### Plasmid construction

The Tri1-Strep constructs used for ectopic expression studies were PCR amplified from genomic *C. trachomatis* L2 (434/Bu) DNA and subcloned into the EcoRI and NotI sites in pcDNA4.0/2xStrepII (55) using In-Fusion cloning (Takara). Tri1-Strep-sfGFP was generated by subcloning superfolder (sf) GFP (from Dre1-sfGFP) into Tri1-Strep constructs (Full length or $Tri1_{84-147}$) at the Xho1 and Apa1 sites as a C-terminal fusion (29). TRAF7 constructs were amplified from pUC57-TRAF7 (a plasmid expressing full-length TRAF7) and subcloned into the HindIII and KpnI sites in pmCherry C1 and pcDNA4.0/3X FLAG vector using In-Fusion cloning. All TRAF7 constructs are tagged at the N-terminus, and all Tri1 constructs are tagged at the C-terminus. Wild-type Myc-MEKK3 (in pcDNA3.1 vector) was generated from a kinase-dead mutant (K391A, Addgene plasmid # 44157) (56) using QuikChange Lightning Site-Directed Mutagenesis (Agilent). Only wild-type Myc-MEKK3 was used for experiments. Constructs were verified by forward and reverse sequencing.

## Affinity purifications

For Strep-Tactin affinity purifications in the context of transfections, HEK293T ($6 \times 10^6$ cells per plate) was seeded in one to three 10 cm$^2$ plates. For FLAG affinity purifications, HeLa cells ($3 \times 10^5$ cells) were seeded in each well of two six-well plates. Cells were transfected with the indicated constructs using Continuum Transfection Reagent (GeminiBio), following the manufacturer's instructions. At 48 h post-transfection, cells were scraped on ice, pelleted, washed with phosphate buffered saline (PBS), and lysed in 1 mL cold wash buffer (50 mM Tris HCl [pH 7.5], 150 mM NaCl, and 1 mM EDTA) plus 0.5% IGEPAL (Sigma), Complete protease (Roche), and PhosSTOP phosphatase inhibitor (Roche). Lysates were added to 60 µL of FLAG beads (Sigma) or 60 µL of Strep-Tactin Sepharose beads (IBA) and incubated with rotation overnight at 4°C. Beads were washed three times in wash buffer plus 0.05% IGEPAL and finally once in wash buffer without detergent. Samples were eluted in 45 µL of FLAG peptide (300 µg/mL, Millipore Sigma) in wash buffer plus 0.05% RapiGest (Waters Corp) for FLAG APs or in 45 µL of 10 mM D-desthiobiotin (IBA) in final wash buffer (50 mM Tris-HCl [pH 7.5], 150 mM NaCl, and 1mM EDTA) for Strep APs.

For FLAG APs in the context of L2 infections, HeLa cells were grown to 70% confluency ($0.84 \times 10^6$ cells/well) in three six-well plates per condition. Cells were transfected (as described above) with mCh-TRAF7 for 24 h prior to infection with L2+pTri1$_{FLAG}$ or L2+pIncG$_{FLAG}$ at a multiplicity of infection (MOI) of 5. L2 strains were suspended in MEM supplemented with 10% FBS and centrifuged at 1,000 revolutions per minute (RPM) for 30 min at room temperature onto HeLa cells grown on coverslips. Infected cells were incubated at 37°C in 5% $CO_2$ for 1 h. Infection media were aspirated off, fresh media containing no aTc or 200 ng/mL aTc were added, and cells were incubated at 37°C in 5% $CO_2$ for 24 h. Cells were treated with 10 µM MG132 (Cayman Chemicals) 4 h prior to lysis and processed for FLAG APs as described above.

For co-APs for MS to determine the interactome of TRAF7$_{WD40}$, HEK293T cells were seeded at a density of $6 \times 10^6$ cells per 10 cm plate for next-day transfection. Two plates were used per transfection condition. Cells were transfected with FLAG-TRAF7$_{WD40}$ and Tri1-Strep using a Continuum transfection reagent (Gemini Bio) or Effectene transfection reagent (Qiagen). For control conditions, cells were transfected with empty FLAG-vector and Tri1-Strep (full length or Tri1$_{1-128}$-Strep). At 44 h post-transfection, cells were treated with 10 µM MG132. At 48 h post-transfection, cells were scraped on ice, pelleted, washed with PBS, lysed in 1 mL cold wash buffer, and subjected to FLAG APs as described above.

To determine the interactome of Tri1$_{FLAG}$, eight six-well plates of 80% confluent HeLa cells were infected with either *C. trachomatis* L2 expressing plasmid-encoded Incs (L2+pTri1$_{FLAG}$, L2+pIncE$_{FLAG}$, or L2+pDre1$_{FLAG}$) or empty vector at an MOI of 5 for 36 h as previously described (29). Ten micromolar MG132 was added 4 h prior to lysis, and cells were lysed on the plates for 30 min at 4°C in lysis buffer (50 mM Tris-HCl [pH 7.5], 150 mM NaCl, 1 mM EDTA, 0.5% NP-40, PhosStop, and Roche Complete Protease Inhibitor). Lysates were clarified by centrifugation at 13,000 RPM at 4°C for 15 min. Supernatants were then incubated with 30 µL anti-FLAG magnetic beads (Millipore Sigma) rotating overnight at 4°C. Beads were washed three times in wash buffer (50 mM Tris-HCl [pH 7.5], 150 mM NaCl, 1 mM EDTA, and 0.05% NP-40) and then once in final wash buffer. Samples were eluted in 45 µL of elution buffer (100 µg/mL FLAG peptide (Millipore Sigma) in final wash buffer for 25 min at room temperature with continuous gentle agitation. All AP-MS experiments were performed in triplicate and assessed by immunoblotting with FLAG antibody using enhanced chemiluminescence (Amersham Biosciences) and by silver stain (Pierce).

## MS sample preparation

A solution of 10 µL of 8 M urea, 250 mM Tris, and 5 mM DTT (dithiothreiotol) was added to the eluate to achieve a final concentration of ~1.7 M urea, 50 mM Tris, and 1 mM DTT. Samples were incubated at 60°C for 15 min and allowed to cool to room temperature. Iodoacetamide was added to a final concentration of 3 mM and incubated at room

temperature for 45 min in the dark. DTT was then added to a final concentration of 3 mM before adding 1 µg of sequencing-grade trypsin (Promega) and incubating at 37°C overnight. Digested peptide samples were acidified to 0.5% trifluoroacetic acid (TFA) (pH < 2) with 10% TFA stock and desalted using C18 ultra micro spin columns (The Nest Group).

## MS data acquisition and analysis

For AP-MS experiments, samples were resuspended in 15 µL of MS loading buffer (4% formic acid and 2% acetonitrile), and 2 µL was separated by a reversed-phase gradient over a nanoflow 75 µm ID × 25 cm long picotip column packed with 1.9 µM C18 particles (Dr. Maisch). Peptides were directly injected over the course of a 75-min acquisition into a Q-Exactive Plus mass spectrometer (Thermo), and data were acquired in a data-dependent acquisition mode. Raw MS data were searched against the uniprot canonical isoforms of the human proteome (downloaded on 28 February 2020) and the *C. trachomatis* L2 Strain434BuATCCVR-902B proteome, using the default settings in MaxQuant (version 1.6.12.0), with a 0.7 min match-between-runs time window enabled (57). Peptides and proteins were filtered to a 1% false discovery rate in MaxQuant, and identified proteins were then subjected to protein-protein interaction scoring. For PPI scoring, protein spectral counts as determined by MaxQuant search results were used for PPI confidence scoring by SAINTexpress (version 3.6.1) (30). For the TRAF7$_{WD40}$ data analysis, cells co-transfected with empty FLAG vector and either Tri1-Strep or Tri1$_{1-128}$-Strep were used as negative controls. For the infection data analysis, HeLa cells infected with L2 transformed with a plasmid expressing an empty FLAG vector were used as a negative control. SAINT analysis was conducted on data from nine L2 strains each expressing a different FLAG-tagged Inc.

## Fluorescence microscopy

HeLa cells were seeded ($1.2 \times 10^6$ cells/well, three wells per condition) and grown on acid-treated glass coverslips (Warner Instruments) in 24-well plates. Cells were transfected using Effectene as described above.

For fluorescence microscopy of L2-infected HeLa cells, L2 strains were suspended in MEM supplemented with 10% FBS and centrifuged at 1,000 RPM for 30 min at room temperature onto HeLa cells grown on coverslips. Infections were performed at an MOI of 1. Infected cells were incubated at 37°C in 5% $CO_2$ for 1 h. Infection media were aspirated off, fresh media containing no aTc or 50 ng/mL aTc were added, and cells were incubated at 37°C in 5% $CO_2$ for 24 or 48 h.

Cells were fixed in 4% paraformaldehyde (PFA) in PBS for 15 min at room temperature and then permeabilized in 0.2% Triton X-100 in PBS for 15 min at room temperature. Cells were blocked in PBS containing 1% bovine serum albumin (BSA) for 1 h and then stained with the appropriate primary antibody (overnight) and secondary antibody (1 h) in 1% BSA. Coverslips were mounted on Vectashield mounting media with or without DAPI (Vector Laboratories).

Images were acquired using Yokogawa CSU-X1 spinning disk confocal mounted on a Nikon Eclipse Ti inverted microscope equipped with an Andora Clara digital camera and CFI APO TIRF 60× oil or PLAN APO 40× objective. Single Z slices were acquired for all images shown. Images were acquired by NIS-Elements software 4.10 (Nikon). For each set of experiments, the exposure time for each filter set for all images was identical. Images were processed using FIJI Software.

## Bioinformatics

The coiled-coil domain of Tri1 from was predicted by submitting the full-length protein sequence for Tri1 from *C. trachomatis* L2 (434/Bu) to waggawagga, which includes Multicoil and NCOILS predictions (34). Additionally, the Tri1 coiled-coil domain was previously annotated on Uniprot by COILS (58) (Uniprot ID: A0A0H3MBY0) as residues

118–138. This coiled-coil domain prediction was used for Tri1 mutant construction. Transmembrane domain predictions were conducted using Phobius (59).

## ACKNOWLEDGMENTS

We thank Raphael Valdivia (Duke University), Deborah Dean (UCSF), Mary Weber (University of Iowa), Ted Hackstadt (Rocky Mountain Laboratories), Xin Lin (University of Texas, M.D. Anderson Cancer Center), Dan Rockey (Oregon State University), and Isabelle Derre (University of Virginia) for reagents.

J.E., C.M.H., E.M., K.A.-B., J.S., K.P., D.S., J.R.J., N.K., and C.E. were supported by funding provided by the NIH (R56 AI152526 and RO1 AI122747). C.M.H. was also supported by NIH F31 AI52286, T32 AI 60537, T32 GM007810, and the UCSF Discovery Fellows program. J.S. was also supported by NIH F31 AI133951. K.P. was also supported NIH F32 AI133902.

Conceptualization, C.M.H., E.M., J.S., K.P., C.E., and J.E.; Data curation, D.L.S.; Formal analysis, J.R.J. and D.L.S.; Funding acquisition, C.M.H. and J.E.; Investigation, C.M.H., E.M., K.A.-B., C.E., and J.E.; Methodology, C.M.H., E.M., D.L.S., J.S., K.P., C.E., and J.E.; Project administration, D.S. N.J.K., C.E., and J.E.; Resources, J.R.J., D.S., and M.S.; Supervision, D.S., N.J.K., C.E., and J.E.; Validation, C.M.H., E.M., D.L.S., C.E., and J.E.; Visualization, C.M.H., J.S., K.P., C.E., and J.E.; Writing—original draft, C.M.H., D.S., J.S., K.P., C.E., and J.E.; Writing-revisions and editing, C.M.H., D.S., J.S., K.P., C.E., and J.E.

The Krogan Laboratory has received research support from Vir Biotechnology, F. Hoffmann-La Roche, and Rezo Therapeutics. Nevan Krogan has a financially compensated consulting agreement with Maze Therapeutics. He is the president and is on the Board of Directors of Rezo Therapeutics, and he is a shareholder in Tenaya Therapeutics, Maze Therapeutics, Rezo Therapeutics, GEn1E Lifesciences, and Interline Therapeutics.

## AUTHOR AFFILIATIONS

[1]Department of Medicine, University of California San Francisco, San Francisco, California, USA

[2]Department of Cellular and Molecular Pharmacology, University of California San Francisco, San Francisco, California, USA

[3]Department of Pathology, University of California San Francisco, San Francisco, California, USA

[4]Department of Microbiology and Immunology, University of California San Francisco, San Francisco, California, USA

## AUTHOR ORCIDs

Nevan J. Krogan  http://orcid.org/0000-0003-4902-337X
Cherilyn Elwell  http://orcid.org/0000-0001-7702-3938
Joanne Engel  http://orcid.org/0000-0002-2168-9711

## FUNDING

| Funder | Grant(s) | Author(s) |
| --- | --- | --- |
| HHS | NIH | National Institute of Allergy and Infectious Diseases (NIAID) | R56AI152526 | Danielle L. Swaney |
| | | Joanne Engel |
| | | Clara M. Herrera |
| | | Eleanor McMahon |
| | | Jessica Sherry |
| | | Khavong Pha |
| | | Jeffrey R. Johnson |
| | | Cherilyn Elwell |

| Funder | Grant(s) | Author(s) |
|---|---|---|
| HHS \| NIH \| National Institute of Allergy and Infectious Diseases (NIAID) | RO1AI122747 | Nevan J. Krogan |
| | | Joanne Engel |
| | | Clara M. Herrera |
| | | Eleanor McMahon |
| | | Jessica Sherry |
| | | Khavong Pha |
| | | Jeffrey R. Johnson |
| | | Cherilyn Elwell |
| NIH | F31 AI52286 | Clara M. Herrera |
| NIH | T32 AI 60537 | Clara M. Herrera |
| NIH | T32 GM007810 | Clara M. Herrera |
| UCSF Discovery Fellows program | | Clara M. Herrera |
| NIH | F31 AI133951 | Jessica Sherry |
| NIH | F32 AI133902 | Khavong Pha |

## DATA AVAILABILITY

All AP-MS raw data files and search results are available from the Pride partner ProteomeXchange repository under the PXD049432 identifier (60, 61) .

## ADDITIONAL FILES

The following material is available online.

### Supplemental Material

**Supplemental figures and table (Spectrum00453-24-s0001.pdf).** Fig. S1-S5; Table S1.

### Open Peer Review

**PEER REVIEW HISTORY (review-history.pdf).** An accounting of the reviewer comments and feedback.

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
