## [Reviewer comments · Microbiology Spectrum]

Microbiology Spectrum

The *Chlamydia trachomatis* Inc Tri1 interacts with TRAF7 to displace native TRAF7 interacting partners

Clara Herrera, Eleanor McMahon, Danielle Swaney, Jessica Sherry, Khavong Pha, Kathleen Boone-Adams, Jeffrey Johnson, Nevan Krogan, Meredith Stevers, David Solomon, Cherilyn Elwell, and Joanne Engel

Corresponding Author(s): Joanne Engel, University of California, San Francisco

Review Timeline:

Submission Date:	February 26, 2024
Editorial Decision:	March 19, 2024
Revision Received:	April 23, 2024
Accepted:	April 23, 2024

Editor: Brian Conlon

Reviewer(s): The reviewers have opted to remain anonymous.

Transaction Report:

DOI: <https://doi.org/10.1128/spectrum.00453-24>

Re: Spectrum00453-24 (The *Chlamydia trachomatis* Inc Tri1 interacts with TRAF7 to displace native TRAF7 interacting partners)

Dear Dr. Joanne Engel:

Thank you for the privilege of reviewing your work. Below you will find my comments, instructions from the Spectrum editorial office, and the reviewer comments.

Reviewer 2 has some minor comments that can likely be easily addressed with some minor edits/additions to the text.

Revision Guidelines

Sincerely,
Brian Conlon
Editor
Microbiology Spectrum

Reviewer #1 (Comments for the Author):

This is a highly detailed and very polished manuscript that describes the authors' identification of TRAF7, a human ubiquitin ligase, as an interaction partner of the chlamydial Inc protein, Tri1. They demonstrate the specificity of the association between the proteins at the inclusion membrane during infection and map the relevant interaction sub-domains. The analysis is robust, appropriately controlled and validated with complementary approaches, lending high confidence in the strength and specificity of

the interaction during cellular infection. Although the specific role that TRAF7 engagement by Tri1 may play was not identified, their findings point to engagement of JNK-AP1 signaling with implications for modulation of innate inflammation, a key feature of chlamydial pathogenesis.

Reviewer #2 (Comments for the Author):

An interaction between the *C. trachomatis* inclusion membrane protein CT224 and host TRAF7 was identified in a previous study using an affinity-purification-mass spectroscopy screen. Herrera, et al confirm the interaction herein and perform a basic characterization of the protein-protein interaction. The authors find evidence for domain-specific interactions and designate CT224 as Tri1 (TRAF7 interactor 1). Based on their findings that a Tri1 coil-coil domain interacts with the TRAF7 WD40 domain to prevent TRAF7 interaction with signal transduction kinases, the authors speculate that this could impact immune signaling within infected cells. Novel findings include recruitment of TRAF7 to the chlamydial inclusion, elucidation of interaction domains, and impact of MEKK2/3 binding to TRAF7. This doesn't move the needle very far forward and the biological relevance of the interaction remains unaddressed. Having said that, the experiments/data being presented are solid and support the author's conclusions.

1. I wonder if the designation of CT224 as Tri1 is a bit too broad. I don't have any good alternatives, but there are multiple TRAFs. CT224 seems to interact only with TRAF7. To base the name on the root "TRAF interactor" doesn't seem very specific.
2. Line 94-95 states that TRAF7 is involved in pathways relevant to chlamydial infection. There is no reference provided. Is this speculation or is there experimental evidence supporting this statement? The point is addressed in the discussion, but it might be helpful for some context in the introduction.
3. Was mutagenesis of Tri1 attempted? It would be helpful to know if the gene might be essential if attempts were not successful.
4. There is precedent for the model being described here (Interaction of a host protein with an Inc resulting in interference with signal transduction). *C. pneumoniae* expresses an Inc that interferes with IL-17 signaling by sequestering a signal transduction protein (Act1) at the inclusion membrane (PMID: 19159390). This should probably be mentioned in the discussion to bolster the author's speculation.

We thank the reviewers for their positive and helpful comments and have addressed each of reviewer 2's suggestions and questions.

1. I wonder if the designation of CT224 as Tri1 is a bit too broad. I don't have any good alternatives, but there are multiple TRAFs. CT224 seems to interact only with TRAF7. To base the name on the root "TRAF interactor" doesn't seem very specific.

We appreciate that the effector name Tri1 does not specifically encompass that it interacts specifically with TRAF7 as opposed to the other TRAFs, but we also were unable to come up with a name encapsulated this subtlety.

2. Line 94-95 states that TRAF7 is involved in pathways relevant to chlamydial infection. There is no reference provided. Is this speculation or is there experimental evidence supporting this statement? The point is addressed in the discussion, but it might be helpful for some context in the introduction.

We have added in the appropriate references.

3. Was mutagenesis of Tri1 attempted? It would be helpful to know if the gene might be essential if attempts were not successful.

For the reviewer's information, we spent several years trying to make a Tri1 mutant. Using FRAEM, we were able to reproducibly get integration of the allelic exchange plasmid into the chromosome (ie, meridioids which exhibited both mCherry and GFP markers and were ampicillin resistant) but were never able to recover recombinants in which the integrated plasmid had recombined to leave mCherry and Ampicillin interrupting the Tri1 locus. As a control, in experiments performed at the same time, we could easily recovery strains in which an IncE allelic exchange plasmid successfully recombined into and then out of the native IncE locus. More recently, Ouellette and co-workers have reported successfully using Crispr1 to generate a strain of L2 in which Tri1 mRNA can be depleted. They generously shared this strain with our lab. Their studies, which are posted on BioRxiv (www.biorxiv.org/content/10.1101/2023.10.17.562819v1.full.pdf), and our unpublished studies, have failed to reveal any changes in the ability of this Tri1 depleted strain to propagate in cell culture. We are continuing to characterize this depletion strain and hope that it will be the focus of a future paper.

4. There is precedent for the model being described here (Interaction of a host protein with an Inc resulting in interference with signal transduction). *C. pneumoniae* expresses an Inc that interferes with IL-17 signaling by sequestering a signal transduction protein (Act1) at the inclusion membrane (PMID: 19159390). This should probably be mentioned in the discussion to bolster the author's speculation.

We thank the reviewer for bringing this paper to our attention and have added in a sentence and reference as suggested. "There is a precedent for inhibition of immune signaling in another human-adapted Chlamydia species, C. pneumoniae. Here, the Inc CP036 binds to and redirects the Interleukin 17A receptor- binding protein Act1 to the inclusion, modulating Interleukin 17A signaling."

Re: Spectrum00453-24R1 (The *Chlamydia trachomatis* Inc Tri1 interacts with TRAF7 to displace native TRAF7 interacting partners)

Dear Dr. Joanne Engel:

Your manuscript has been accepted, and I am forwarding it to the ASM production staff for publication. Your paper will first be checked to make sure all elements meet the technical requirements. ASM staff will contact you if anything needs to be revised before copyediting and production can begin. Otherwise, you will be notified when your proofs are ready to be viewed.

Sincerely,
Brian Conlon
Editor
Microbiology Spectrum